# New Developments in Pharmacological Treatment of Obesity and Type 2 Diabetes—Beyond and within GLP-1 Receptor Agonists

**DOI:** 10.3390/biomedicines12061320

**Published:** 2024-06-13

**Authors:** Ferenc Sztanek, László Imre Tóth, Attila Pető, Marcell Hernyák, Ágnes Diószegi, Mariann Harangi

**Affiliations:** 1Division of Metabolism, Department of Internal Medicine, Faculty of Medicine, University of Debrecen, H-4032 Debrecen, Hungary; 2Third Department of Internal Medicine, Semmelweis Hospital of Borsod-Abauj-Zemplen County Central Hospital and University Teaching Hospital, H-3529 Miskolc, Hungary; 3Doctoral School of Health Sciences, University of Debrecen, H-4032 Debrecen, Hungary; 4Institute of Health Studies, Faculty of Health Sciences, University of Debrecen, H-4032 Debrecen, Hungary; 5ELKH-UD Vascular Pathophysiology Research Group 11003, University of Debrecen, H-4032 Debrecen, Hungary

**Keywords:** G-protein-coupled receptors, GLP-1 receptor agonist, GIP receptor agonist, glucagon receptor agonist, obesity, type 2 diabetes

## Abstract

Guidelines for the management of obesity and type 2 diabetes (T2DM) emphasize the importance of lifestyle changes, including a reduced-calorie diet and increased physical activity. However, for many people, these changes can be difficult to maintain over the long term. Medication options are already available to treat obesity, which can help reduce appetite and/or reduce caloric intake. Incretin-based peptides exert their effect through G-protein-coupled receptors, the receptors for glucagon-like peptide-1 (GLP-1) and glucose-dependent insulinotropic polypeptide (GIP), and glucagon peptide hormones are important regulators of insulin secretion and energy metabolism. Understanding the role of intercellular signaling pathways and inflammatory processes is essential for the development of effective pharmacological agents in obesity. GLP-1 receptor agonists have been successfully used, but it is assumed that their effectiveness may be limited by desensitization and downregulation of the target receptor. A growing number of new agents acting on incretin hormones are becoming available for everyday clinical practice, including oral GLP-1 receptor agonists, the dual GLP-1/GIP receptor agonist tirzepatide, and other dual and triple GLP-1/GIP/glucagon receptor agonists, which may show further significant therapeutic potential. This narrative review summarizes the therapeutic effects of different incretin hormones and presents future prospects in the treatment of T2DM and obesity.

## 1. Introduction

The prevalence of diabetes and obesity has been rising globally, posing significant public health challenges. According to an analysis of global data, adult obesity rates have more than doubled in women and nearly tripled in men. In 2022, a total of 879 million adults worldwide were living with obesity. Obesity is responsible for approximately 43% of all cases of type 2 diabetes, with rates as high as 80–85% in some populations [1]. Obesity not only increases the risk of developing type 2 diabetes but also exacerbates complications and related comorbidities, such as cardiovascular disease, cancer, and osteoarthritis, while decreasing life expectancy and increasing health care costs. Significant and sustainable weight loss can be beneficial in the development of type 2 diabetes (T2DM) and related complications, as well as in the occurrence of cardiovascular events [2]. Lifestyle changes, such as appropriate diet and regular exercise, are key to weight loss and optimizing adipose tissue metabolism. Pharmacological options are already available for the treatment of obesity, which can help reduce appetite and/or calorie intake [3]. Understanding the role of intercellular signaling pathways and inflammatory processes occurring in adipose tissue is essential for the development of effective pharmacological agents in the complex mechanism of obesity. A therapeutic approach based on innovation and the discovery of mechanisms of action offers promising opportunities for a deeper understanding of the connections between obesity and T2DM, which may lead to the development of new therapeutic options in addition to lifestyle changes [4].

Guidelines for the management of obesity and T2DM emphasize the importance of lifestyle changes, including a reduced-calorie diet and increased physical activity. A weight loss of at least 5% can delay the onset of T2DM and reduce the occurrence of cardiovascular diseases, while long-term sustained weight loss of 10–15% can even induce T2DM remission and provide numerous other health benefits [3,4]. However, achieving and maintaining significant weight loss is particularly challenging in patients with T2DM. Additionally, rapid weight loss may improve insulin sensitivity, which can lead to further weight loss plateau and repeated weight regain [5]. Therefore, the addition of pharmacological therapies with properties that promote weight loss and reduce blood glucose levels represents a desirable approach in the management of obesity and T2DM [2]. In this regard, several incretin-based peptide therapies have emerged: once-weekly subcutaneous injections of glucagon-like peptide-1 (GLP-1) receptor agonist semaglutide, available also as a once-daily tablet, and the dual agonist tirzepatide, which interacts with both GLP-1 and glucose-dependent insulinotropic polypeptide (GIP) receptors, also administered as a once-weekly subcutaneous injection. Clinical trials concluded in the past decades have confirmed that GLP-1 receptor agonists have proven effective in the management of T2DM, and some formulations are now also used in higher doses for the treatment of obesity [6].

The development of incretin-based therapy for T2DM began in the 1990s, with the widespread use of daily short-acting injectable GLP-1 receptor agonists and oral dipeptidyl peptidase-4 inhibitors occurring in the 2000s. In the following decade, long-acting subcutaneous or daily oral GLP-1 receptor agonists administered weekly emerged [7]. In recent years, new formulations have been developed, and dual and/or triple GLP-1/GIP/glucagon receptor agonists have shown promise in the treatment of T2DM [8]. Figure 1 schematically depicts the incretin-based agents that are currently used in everyday medical practice or that are currently under clinical development [9].

Significant progress has been made in recent years in the development and clinical application of new incretin-based peptide therapies for the treatment of obesity and T2DM. The purpose of this narrative review is to summarize the therapeutic effects of various incretin hormones and to present future prospects in the treatment of T2DM and obesity. In addition, our aim is to briefly review recent pharmacological treatments for obesity and type 2 diabetes and summarize the latest results from clinical trials. We obtained the data for this review from PubMed, Scopus, Web of Science, and ClinicalTrials.gov, using the following keywords: “obesity”, “type 2 diabetes”, “new drug treatment”, “recent developments”, “clinical studies”, “G-protein-coupled receptor”, “glucagon-like peptide-1 receptor agonists”, “glucose-dependent insulinotropic polypeptide receptor agonist”, “glucagon receptor agonist”, “dual agonist”, and “triple agonist.”

## 2. The Importance of Inflammatory Processes and Oxidative Stress in Adipose Tissue

White adipose tissue possesses highly complex functions, with its primary role lying in energy storage. The function of white adipose tissue is often impaired in obese and diabetic individuals, resulting in adipocyte hypertrophy, pathological fat accumulation, and ectopic fat deposition, which can lead to the development of metabolic and cardiovascular diseases. Obesity is associated with increased insulin secretion, leading to insulin resistance and ultimately beta-cell failure, thereby promoting the development of T2DM [10].

Numerous inflammatory mediators are produced in adipose tissue, significantly contributing to the development of chronic inflammatory conditions [11] (Figure 2). One of the main characteristics in white adipose tissue is an increase in the number of macrophages, which also participate in the development of insulin resistance [12]. Among obese individuals, it is possible for the phenotype of macrophages to change, indicating the dominance of proinflammatory M1-polarized macrophages over M2 macrophages involved in anti-inflammatory and reparative mechanisms. This suggests a change in the innate immune system in obesity [13]. Due to chronic inflammation, the secretion profile of adipokines in white adipose tissue also shifts, increasing the production of proinflammatory factors such as tumor necrosis factor-alpha (TNFα), interleukin-1 (IL-1), interleukin-6 (IL-6), interleukin-8 (IL-8), and leptin, while decreasing the secretion of anti-inflammatory interleukin-10 (IL-10) and adiponectin. M1-polarized macrophages enhance the development of abnormal lipid profiles and insulin resistance, regardless of the inflammatory mediators in adipose tissue. Ultimately, persistent inflammation occurs in dysfunctional adipose tissue due to hypoxia, fibrosis, and mitochondrial dysfunction, leading to systemic inflammation, progression of atherosclerosis, and the development of T2DM and cardiovascular diseases [14].

Obesity and associated insulin resistance are accompanied by increased oxidative stress in endothelial cells, during which the production of reactive oxygen species (ROS) surpasses the capacity of antioxidant factors to balance them. As a result, signaling pathways are modified, and the lipid, protein, and DNA content of cells is damaged. The superoxide anion, with high reactivity, binds and inactivates the physiological function of endothelial nitric oxide synthase (eNOS), leading to the production of peroxynitrite instead of NO. Peroxynitrite is a potent oxidizing agent that can cause lipid peroxidation, protein tyrosine nitration, DNA damage, and cell death. Additionally, peroxynitrite oxidizes tetrahydrobiopterin, an essential cofactor for eNOS, rendering it biologically inactive and unable to catalyze endothelial NO synthesis [15]. Consequently, eNOS catalyzes the production of further superoxide radicals instead of NO. Beyond other intracellular signaling and metabolic changes, the inactivation of eNOS is a key factor in the relationship between oxidative stress and pathological functional changes in the endothelium. In diabetes, chronic hyperglycemia, acute fluctuations in blood glucose levels, and insulin resistance all contribute to increased oxidative stress and, therefore, to the development of endothelial dysfunction [16].

GLP-1 receptor agonists may potentially reduce inflammation and oxidative stress in adipose tissue by affecting the expression of cytokines, chemokines, and macrophage infiltration, with prolonged GLP-1 action potentially alleviating obesity-related adipose tissue inflammation while enhancing the expression of anti-inflammatory genes [17,18] (Figure 2).

## 3. The Importance of G-Protein-Coupled Receptors in Obesity and T2DM

Incretin-based peptides exert their cellular effects through G-protein-coupled receptors, which have become important therapeutic targets in the treatment of obesity and T2DM in recent years. However, the exact mechanism underlying G-protein-coupled receptors/G-protein specificity remains ambiguous and warrants further investigation [19]. G-protein-coupled receptors constitute the largest family of membrane proteins, and ligand binding can lead to activation of multiple intracellular pathways, including G proteins, beta-arrestins, and other non-G-protein-regulated signaling pathways, such as mitogen-activated protein kinase phosphorylation [20] (Figure 3). Molecules that bind directly to G-protein-coupled receptors can be either orthosteric (binding to the same pocket as the natural ligand) or allosteric (binding to a different pocket). Allosteric modulators can cooperatively interact with the natural orthosteric ligand, thereby enhancing or attenuating receptor signaling. Beta-arrestins contribute to a reduction in the response to receptor agonism through internalization of the specific G-protein-coupled receptor [21]. Presumably, the effect of native GLP-1 on the GLP-1 receptor evenly activates various signaling pathways. However, certain agonists of activated G-protein-coupled receptors can promote the activation of a subset of signaling pathways, resulting in “ligand bias” associated with unique cellular responses. Deviation from the typical effect profile of a certain receptor type is termed “biased agonism.” Certain biased agonists of the GLP-1 receptor may more strongly facilitate G-protein activation than recruitment of beta-arrestins [22]. However, it is possible that bias towards G-protein signaling may stem more from reduced activation of G-protein-coupled receptor kinases than inhibition of binding to beta-arrestins, while other studies have shown that certain β2AR ligands prefer beta-arrestin signaling over G-protein activation [23,24]. Beta-arrestin-dependent seven-transmembrane receptor signaling includes beta-arrestins, which function as multifunctional proteins regulating receptor desensitization, endocytosis, and ubiquitination while modulating G-protein signaling and promoting activation of alternative intracellular pathways. G-protein-based GLP-1 agonists appear to achieve significant weight and blood sugar-reducing effects, in part through reduced beta-arrestin recruitment, thereby avoiding receptor desensitization and downregulation [25]. β-arrestins intracellularly exert an inhibitory effect on signaling pathways mediated by G-protein-coupled receptors; concurrently, they possess the capability to regulate cellular functions independently of G proteins, such as through interaction with peroxisome proliferator-activated receptor-γ [26]. It appears that both G-protein signal activation and the beta-arrestin pathway may facilitate weight loss by inhibiting adipogenesis and inflammatory response [23,27]. Different intracellular cellular responses following binding to G-protein-coupled receptors may also explain the different efficacy of different peptide-based therapies (Figure 3).

GLP-1, GIP, and glucagon receptors belong to class B G-protein-coupled receptors within the secretin receptor family. Their activation involves G protein (Gαs), leading to increased insulin secretion in beta cells via cyclic adenosine monophosphate (cAMP) production, but they also activate the beta-arrestin pathway, which impedes G-protein signaling. Each receptor undergoes agonist-induced endocytosis to varying degrees, thus fine-tuning spatial and temporal modulation of intracellular signaling and hormonal response [28]. During biased agonism, the balance between receptor recruitment, activation of intracellular signaling, and subsequent receptor trafficking may be ligand specific. There are numerous examples of biased agonism at the GLP-1 receptor, including both natural and pharmacological orthosteric agonists. Biased GLP-1 receptor agonists derived from exendin-4 result in prolonged insulin secretion by reducing GLP-1 receptor desensitization and endocytosis, as well as mitigating GLP-1 receptor downregulation [29]. The dual GLP-1/GIP receptor agonist tirzepatide exhibits unbalanced dual agonism at GLP-1 and GIP receptors, but with significantly lower affinity for the GLP-1 receptor and much milder cAMP activation [30].

GLP-1, GIP, and glucagon receptors are present on pancreatic beta cells, and their physiological or pharmacological activation leads to significant metabolic effects [9] (Figure 4). Activation of all three receptors enhances glucose-stimulated insulin secretion, but they differ in their effects on fasting and postprandial blood glucose levels, gluconeogenesis, and lipogenesis, while their common effects include increased energy expenditure and enhanced satiety for weight loss [31]. GLP-1 receptor agonists moderate fasting and postprandial blood glucose levels and, beyond their weight-reducing effects, have demonstrated anti-atherogenic protection in terms of cardiovascular diseases [32,33]. In the case of GIP analogs, beyond effective blood sugar reduction, they have also shown effects on food intake and weight loss, especially when combined with GLP-1 receptor agonists. However, the effects of GIP on fatty acid metabolism, atherosclerosis, adipose tissue metabolic activity, and energy expenditure remain unclear [33]. Despite glucagon’s blood sugar-increasing effect, glucagon receptor activation moderates food intake, enhances fatty acid breakdown, and increases energy expenditure, supporting the rational use of glucagon in combination with other incretin hormones in the treatment of obesity and T2DM [34].

### 3.1. The Main Effects of Glucagon-like Peptide-1

Glucagon-like peptide-1 hormone (GLP-1) is produced in the L cells of the small intestine following meals and stimulates insulin secretion in a glucose-dependent manner. Its action activates the GLP-1 receptor, a glycoprotein receptor belonging to the family of G-protein-coupled receptors with seven transmembrane helices [35]. Activation of GLP-1 receptors leads to the formation of cAMP from adenosine triphosphate (ATP). cAMP activates protein kinase A, resulting in closure of ATP-sensitive potassium channels and depolarization of beta cells, leading to the opening of voltage-gated calcium channels on the cell membrane and exocytosis of insulin-containing granules released from the endoplasmic reticulum due to calcium release [36]. Additionally, experimental studies suggest that activation of GLP-1 receptors inhibits beta-cell apoptosis [37]. Metabolic effects mediated by GLP-1 include increased glucose uptake and glycogen production in skeletal muscle, as well as inhibition of gluconeogenesis in the liver [38] (Figure 4). Due to its positive chronotropic and inotropic effects on cardiac muscle cells, GLP-1 slightly increases the heart rate, which is counteracted by the favorable effect of incretin hormones on left ventricular diastolic dysfunction, common in T2DM [39,40]. GLP-1 inhibits glucagon secretion in alpha cells, slows gastric emptying, reduces appetite, and increases satiety through GLP-1 receptors in the hypothalamus in the central nervous system. In cardiovascular protection, the anti-atherogenic, anti-inflammatory effect on the vascular wall and inhibition of macrophage activation play a significant role [18,41]. Animal models have shown that GLP-1 can increase energy expenditure and regulate heat production and conversion from white adipose tissue to brown adipose tissue in obese mice [42], although human data on this effect are not yet convincing [18,43]. The weight-reducing effect achieved with GLP-1 receptor agonists may alleviate cardiac insulin resistance in obese and overweight individuals. Reduced insulin resistance may be linked to decreased ejection fraction in individuals with T2DM [44] and in non-diabetic obese patients [45]. These effects may partially contribute to the cardioprotective effects of agents targeting the incretin axis, particularly in the context of heart failure. Furthermore, GLP-1 receptor agonists have been demonstrated to exert a protective effect on the microvascular compartment of the myocardium [46,47]. Substantial evidence also supports the beneficial effects of the GLP-1 hormone on epicardial adipose tissue [48] and in the treatment of metabolic-associated fatty liver disease (MAFLD—formerly known as non-alcoholic fatty liver disease) [49]. Overall, the main effects of GLP-1 include reducing fasting and postprandial blood glucose levels, weight loss, and confirmed anti-atherogenic protection against cardiovascular diseases.

### 3.2. The Main Effects of Glucose-Dependent Insulinotropic Polypeptide

Glucose-dependent insulinotropic polypeptide (GIP) is produced from K cells in the mucosa of the small intestine and exerts its effects through a receptor coupled to a G protein. The glucose-dependent insulin-releasing effect of GIP, like GLP-1, is impaired in T2DM, but combining them may further improve beta-cell function and insulin secretion [50] (Figure 4). Similar to GLP-1, GIP also inhibits beta-cell apoptosis in the pancreas, thereby improving its survival during T2DM progression [51]. GIP also stimulates bone formation by inhibiting osteoclast apoptosis [52]. The precise effects of GIP on lipid metabolism are currently unclear; some studies suggest it may increase lipid uptake in adipose tissue and decrease lipolysis [53,54], while others suggest GIP may have lipolytic activity, as adipose tissue from transgenic mice overexpressing GIP has been found to be resistant to high-fat feeding [55,56]. It is hypothesized that by enhancing lipid-buffering capacity and insulin sensitivity in white adipose tissue postprandially, GIP may prevent ectopic fat deposition between tissues and muscles [57]. Activation of GIP receptors improved blood glucose levels in obese mouse models without altering body weight or fat tissue quantity [58], and activation of GIP receptors in the hypothalamus reduced food intake as well [59]. Whole-genome association studies have shown associations between different activity levels of human GIP receptor variants and the degree of obesity [60]. The anti-inflammatory effect of GIP on atherosclerosis is not clear from preclinical studies; therefore, the role of GIP in the atherosclerosis process seems to be controversial at the moment. In cell cultures, the administration of GIP may exert both anti-atherogenic effects (increased nitric oxide and adiponectin concentration, decreased endothelin-1 levels, reduced oxidative stress, decreased migration and proliferation of fibroblast cells) and pro-atherogenic effects (decreased adiponectin levels, increased endothelin-1 and osteopontin levels) [61]. In another study, high-dose administration of GIP reduced macrophage translocation into the vessel wall, thereby reducing foam cell formation, decreasing matrix metallopeptidase-9 activity and interleukin-6 secretion, and ultimately reducing the size of atherosclerotic plaques [62]. In summary, besides effectively reducing blood sugar levels, GIP analog treatment has also been shown to affect food intake and weight loss, especially when combined with GLP-1 receptor agonists. However, the detailed effects of GIP on lipid metabolism, atherosclerosis, metabolic activity of adipose tissue, and energy expenditure are still unclear.

### 3.3. The Main Effects of Glucagon

Glucagon is released from pancreatic α-cells through cleavage of its precursor, proglucagon. In maintaining blood glucose levels, glucagon responds to low glucose levels by increasing gluconeogenesis and glycogenolysis in the liver (Figure 4). In T2DM, the inhibition of glucagon secretion in response to serum glucose levels is impaired, leading to sustained hyperglycemia due to prolonged hormone action [63]. Similar to GLP-1 and GIP, glucagon stimulates beta cells; however, it acts differently on alpha cells: While GLP-1 continuously inhibits glucagon secretion, GIP stimulates glucagon secretion under normal blood sugar and hypoglycemic conditions, but not during hyperglycemia. This physiological effect may mitigate postprandial hyperglycemia and provide protection against the adverse effects of persistently high blood sugar levels [64]. Glucagon exerts its effects through the glucagon receptor, which is also a seven-transmembrane receptor coupled to G proteins. Glucagon receptors are primarily present in liver cells but can also be found in the central nervous system, kidneys, digestive system, and pancreas [65]. In addition to its blood glucose-increasing effect, glucagon activates fat breakdown and inhibits lipogenesis in the liver, reducing concentrations of non-esterified fatty acids and triacylglycerol, and moderating triglyceride content in liver cells in animal experiments [66]. Based on these findings, activation of the glucagon receptor may have a favorable effect on the development and treatment of MAFLD, commonly seen in T2DM. Glucagon receptors are also expressed in pancreatic beta cells, showing significant amino acid homology with incretin receptors and promoting insulin secretion. Glucagon enhances the feeling of satiety and exerts an anorexigenic effect by crossing the blood–brain barrier and acting on the hypothalamus [67]. In animal models, glucagon stimulates energy expenditure through activation of brown adipose tissue [68], although the significance of this effect has not been confirmed by human data [69]. In summary, despite its blood glucose-increasing effect, glucagon moderates food intake, enhances fatty acid breakdown, and increases energy expenditure. These properties support the rational use of glucagon in combination with other incretin hormones in the treatment of T2DM.

## 4. Peptide-Based Therapies in the Treatment of Obesity

The GLP-1 receptor agonist liraglutide at a high dose (administered as a single daily injection of 3 mg subcutaneously) has been approved as an adjunctive medication for weight loss achieved through a low-calorie diet and physical activity in adults with obesity or overweight-related comorbidities, as well as for the treatment of obese children and adolescents aged 12 years or older [70]. Previous large randomized clinical trials have demonstrated that a 3 mg dose of liraglutide can result in an average weight loss of 5.6 kg over nearly one year in overweight or obese individuals without diabetes [71]. The results achieved with high-dose liraglutide have encouraged further development of peptide therapies for the management of obesity and overweight. Recent significant clinical trials evaluating the efficacy of weight loss are summarized in Figure 5 and Figure 6.

The STEP clinical trials collectively provide valuable information about semaglutide’s safety, efficacy, and impact on weight loss and cardiovascular outcomes in various patient populations. The STEP trials have evaluated the effects of once-weekly subcutaneous administration of the GLP-1 receptor agonist semaglutide at a dose of 2.4 mg [72] (Figure 5). Generally, approximately 15% weight loss was observed in overweight and obese, non-diabetic adults across various treatment periods, lasting up to 2 years (from STEP-1 to STEP-6, STEP-8) [72,73,74,75,76,77,78,79]. Similar efficacy was achieved with once-weekly subcutaneous injections of 2.4 mg semaglutide in the STEP Teens trial among adolescents aged 12–18 years [80], as well as with high-dose oral administration of 50 mg semaglutide in overweight and obese, non-diabetic adults (OASIS-1) [81] (Figure 6). In the STEP-2 trial, a smaller but significant 9.6% weight loss was observed with weekly subcutaneous administration of 2.4 mg semaglutide in obese patients with T2DM [73]. In the 52-week STEP-HFpEF study, besides significant weight reduction in obese patients with heart failure with preserved ejection fraction treated with once-weekly subcutaneous 2.4 mg semaglutide, a decrease in heart failure symptoms (fatigue, dyspnea, and edema) and improvement in physical performance were observed [82]. In the SELECT trial, over an average of 40 weeks of follow-up, a 20% reduction in the occurrence of the three-point composite cardiovascular endpoint (cardiovascular death, nonfatal myocardial infarction, and nonfatal stroke) was found in overweight and obese, non-diabetic adults with once-weekly subcutaneous semaglutide (2.4 mg) [83].

In clinical trials with high-dose semaglutide, nausea and vomiting were the most common side effects, which were associated with delayed gastric emptying and occurred in approximately 40% of participants in the studies. These side effects were generally mild to moderate in severity and resolved after 1–3 months of treatment, leading to treatment discontinuation in less than 5% of cases. Initial gastrointestinal side effects were observed in approximately 80% of participants taking high-dose oral semaglutide (50 mg once daily), but like with subcutaneous weekly 2.4 mg semaglutide, their severity decreased over time [81]. Currently, high-dose subcutaneous semaglutide (2.4 mg once weekly) is approved in the United States and Europe for the treatment of obesity and overweight as an adjunct to a calorie-restricted diet and increased physical activity. Further data are needed for regulatory approval as part of the ongoing OASIS trial program evaluation.

The design of the SURMOUNT clinical trials of experiments with the dual GLP-1 and GIP receptor agonist tirzepatide bears many similarities to the STEP program (Figure 6). The SURMOUNT trials have demonstrated tirzepatide glucose-lowering efficacy, weight loss efficacy, and cardiovascular safety in various patient populations. In the SURMOUNT-1 and -3 studies, tirzepatide administered at a high dose of 15 mg subcutaneously once weekly resulted in approximately a 20% reduction in body weight in overweight and obese, non-diabetic adults, even after up to 2 years of treatment [84,85]. Weight loss was slightly lower, at 14.7%, in overweight and obese patients with T2DM in the SURMOUNT-2 study [86]. To minimize gastrointestinal side effects, the tirzepatide dose was gradually increased in all SURMOUNT clinical trials, starting at 2.5 mg and escalating by 2.5 mg every 4 weeks until reaching the maintenance dose required. Initial gastrointestinal side effects were reported by approximately 30% of participants, which were generally mild to moderate in severity and significantly diminished during sustained treatment achieved through dose titration, resulting in a discontinuation rate of less than 8% [87]. Several large, randomized clinical trials are currently underway in the SURMOUNT program, aiming to determine the effectiveness of tirzepatide in promoting weight loss in overweight and obese children, individuals with obstructive sleep apnea, and those with heart failure with preserved ejection fraction [88]. A longer-term study, SURPASS-CVOT, is ongoing to evaluate the effect of tirzepatide on major cardiovascular events in overweight and obese adults, with an expected completion date of October 2024 [88]. Tirzepatide has been approved by the United States Food and Drug Administration (FDA) and the European Medicines Agency (EMA) for the treatment of weight in obese and overweight individuals.

High-dose semaglutide and tirzepatide have been extensively studied for their efficacy in weight management and glycemic control. However, their use can be associated with several side effects, which are important to consider during clinical application. Gastrointestinal symptoms such as nausea, vomiting, diarrhea, or constipation were the most common side effects, observed at a significantly higher incidence in patients receiving high-dose semaglutide or tirzepatide compared to comparators. These side effects are usually mild to moderate in severity and tend to be temporary. The use of semaglutide and tirzepatide may increase the risk of biliary disorders such as cholelithiasis, a risk similar to that of other GLP-1 receptor agonists, and therefore patients should be monitored during clinical use [89,90]. There is concern about the potential risk of pancreatitis and pancreatic and thyroid cancers with the use of GLP-1 receptor agonists. However, the incidence of these conditions is low, and definitive conclusions cannot be drawn at this time. In clinical trials, severe hypoglycemia, acute pancreatitis, gallstones, and cholecystitis occurred rarely and were observed only in a few cases during the use of high-dose semaglutide and tirzepatide [89,90].

## 5. Peptide-Based Therapies in the Treatment of Obesity and Type 2 Diabetes

### 5.1. Dual GIP/GLP-1 Receptor Agonists

The first GIP/GLP-1 receptor co-agonist was discovered in 2013, exhibiting significantly more pronounced glucose-lowering and insulin-releasing efficacy than selective GLP-1 receptor agonists in obese and leptin receptor-deficient mouse models, as well as in humans [91]. Further development of GIP/GLP-1 receptor co-agonists followed these results, with tirzepatide becoming the most widely studied representative of this drug class. Tirzepatide binds more strongly to the GIP receptor than to the GLP-1 receptor; in receptor-binding studies, tirzepatide’s affinity was comparable to the native hormone’s affinity for the GIP receptor, while it was approximately five times weaker than native GLP-1’s affinity for the receptor [92]. However, binding to the GLP-1 receptor also seems significant in tirzepatide’s cell surface receptor mechanism, as tirzepatide had no effect on body weight, food intake, fasting insulin levels, skeletal muscle, subcutaneous adipose tissue endogenous glucose production, or insulin-stimulated glucose uptake in GLP-1 receptor non-expressing knockout mice [93].

The US and European drug regulatory agencies approved tirzepatide for the treatment of T2DM in 2022. As part of the SURPASS program, phase III trials were conducted to evaluate the therapeutic efficacy (glycemic control and weight reduction), safety, and tolerability of tirzepatide in patients with T2DM [94,95]. Tirzepatide treatment was administered in three doses (5 mg, 10 mg, or 15 mg weekly subcutaneously), starting with a 2.5 mg weekly dose, then titrated up every 4 weeks. These studies show that tirzepatide significantly reduced fasting blood glucose and HbA1c levels not only compared to placebo but also compared to degludec and glargine insulins or semaglutide. The change in HbA1c ranged from −1.87% to −2.59% for various doses of tirzepatide compared to placebo, −1.86% for weekly 1 mg semaglutide in the SURPASS-2 trial, −1.34% for degludec insulin in the SURPASS-3 trial, and −1.44% for glargine insulin in the SURPASS-4 trial [94,95,96] (Figure 7). Additionally, tirzepatide significantly reduced weight in obese T2DM patients in a dose-dependent manner; by weeks 40–52 of the trials, the 5 mg dose of tirzepatide resulted in a weight loss of 6.2–7.8 kg, the 10 mg dose led to a weight loss of 7.8–10.7 kg, and the 15 mg dose caused a weight loss of 9.5–12.9 kg [96]. Currently, several dual GIP/GLP-1 receptor agonists are under clinical development, among which tirzepatide was approved in the United States and Europe in 2022 for the treatment of T2DM.

### 5.2. Dual Glucagon/GLP-1 Receptor Agonists

Glucagon alters lipid metabolism and energy expenditure favorably while also reducing appetite and food intake, making it an attractive option for the treatment of T2DM and obesity, especially when combined with the insulinotropic effects of GLP-1. The first monomolecular glucagon receptor/GLP-1 receptor co-agonist was discovered in 2009, and its weekly administration significantly improved carbohydrate metabolism and reduced weight in obese mice fed a high-calorie diet during preclinical studies [97]. In recent years, two glucagon receptor/GLP-1 receptor co-agonists underwent phase II clinical trials in human experiments, namely, SAR425899 and cotadutide [98,99]. Further development of SAR425899 was eventually halted during human trials. Cotadutide effectively lowered blood sugar levels and reduced weight in obese individuals and T2DM patients, increased insulin secretion, delayed gastric emptying, with the most common side effects being gastrointestinal symptoms (nausea and vomiting), which were dose-dependent [99,100]. A multicenter study evaluated the effects of subcutaneously administered cotadutide in 834 participants in daily doses of 100 μg, 200 μg, and 300 μg compared with placebo and 1.8 mg liraglutide daily [101]. The study lasted for 54 weeks and assessed liver abnormalities and metabolic parameters in overweight and obese T2DM patients. Every dose of cotadutide and liraglutide significantly reduced HbA1c levels compared to placebo (change in HbA1c—cotadutide 100 μg: −1.03%; 200 μg: −1.16%; 300 μg: −1.19%, liraglutide 1.8 mg: −1.17%); however, there was no significant difference in HbA1c reduction between cotadutide and liraglutide. The highest dose of cotadutide resulted in a significantly greater reduction in weight compared to liraglutide and placebo (change in weight—cotadutide 100 μg: −3.7 kg; 200 μg: −3.22 kg; 300 μg: −5.02 kg, liraglutide 1.8 mg: −3.33 kg). Higher doses of cotadutide also showed more significant improvement in liver enzyme levels and parameters indicative of liver fibrosis compared to liraglutide [101]. Based on the favorable results, the effects of cotadutide are primarily planned to be extended towards the treatment of MAFLD. A comparison of the HbA1c and weight-reducing effects of cotadutide with various doses of dulaglutide, liraglutide, and semaglutide currently in clinical use is shown in Figure 3 [102].

### 5.3. Triple GIP/GLP-1/Glucagon Receptor Agonists

The significant results of clinical trials with dual receptor agonists have stimulated the development of new combination approaches in the treatment of T2DM and obesity. The synergistic metabolic benefits of pharmacological modulation of GLP-1, GIP, and glucagon receptors were first demonstrated in 2015. Triple agonism of GLP-1, GIP, and glucagon resulted in greater weight loss and reduction in blood glucose levels in obese mice compared to monoagonists and dual GLP-1/GIP receptor agonists [31]. A triple receptor agonist, known as SAR441255, dose-dependently reduced weight by up to 25% after 4 weeks of treatment. When administered subcutaneously, SAR441255 at the two highest test doses (80 µg and 150 µg) normalized fasting blood glucose levels within the first hour of treatment in overweight individuals, with the highest dose being well tolerated and the most common treatment-related side effects being gastrointestinal complaints [103]. New triple receptor agonists with longer duration of action have been developed, allowing for once-weekly dosing. In a recent study, these novel, long-acting triple receptor agonist formulations more effectively and successfully reduced weight in obese mouse models compared to semaglutide, tirzepatide, dual GLP-1/glucagon receptor agonists, or short-acting triple receptor agonists, and their use did not entail increased risk of hypoglycemia [104]. In a recent phase I study, retatrutide, a triple glucagon/GIP/GLP-1 receptor agonist, significantly reduced body weight and improved glycemic control, exhibiting a safety and tolerability profile similar to other incretin formulations in study participants, and body weight continued to decrease steadily after a single dose until the end of the study [105] (Figure 6). Retatrutide has also been subjected to human trials in patients with T2DM, showing significant reductions in body weight and HbA1c by the end of the 12th week compared to baseline values [106] (Figure 7). Whether triple receptor agonists will surpass the clinical outcomes of GLP-1/GIP receptor agonists will be determined by further investigations in the future.

## 6. Studies with Other Weight-Loss Medications

Orforglipron is an orally administered, non-peptide GLP-1 receptor agonist that can be taken once daily. However, it binds differently to the GLP-1 receptor compared to native GLP-1, thereby inducing G-protein activation rather than beta-arrestin recruitment [107]. It is being investigated for the treatment of obesity and T2DM and could represent a competitive alternative to oral semaglutide, as its less burdensome administration does not require fasting. In T2DM patients, the average change in HbA1c was −2.1% with a dose of 45 mg of orforglipron (compared to dulaglutide and placebo), and nearly half of the participants achieved more than a 10% reduction in body weight in a 26-week phase 2 trial [108]. In obese, non-diabetic individuals, orforglipron administration for 36 weeks in a phase 2 trial, ranging from 12 mg to 45 mg doses, resulted in dose-dependent weight loss of up to 14.7%, accompanied by improvements in cardiometabolic risk factors [109] (Figure 6).

Danuglipron is another orally administered, non-peptide GLP-1 receptor agonist oriented towards G-protein [110]. In a phase 2 study in T2DM patients over 16 weeks, the highest dose of danuglipron administered twice daily (120 mg) resulted in an average HbA1c reduction of 1.2%, with an average weight loss of 4.2 kg [111]. A recently completed phase 2 trial in obese and non-diabetic individuals revealed, according to a press release, that danuglipron at doses ranging from 40 to 200 mg twice daily resulted in a 11.7% weight loss after 32 weeks of treatment compared to placebo. However, the rate of discontinuation due to adverse events (gastrointestinal side effects) exceeded 50% in each dosage group compared to placebo [112].

Several potential formulations for obesity treatment are currently undergoing clinical trials. The triple GLP-1/GIP/glucagon receptor agonist retatrutide (administered once weekly as 12 mg subcutaneous injections) achieved a 24% weight loss in a 48-week study involving overweight and obese individuals [113]. Several unimolecular dual or triple agonist proteins are also currently being studied, including those based on the action of the oxyntomodulin hormone, such as survodutide, pemvidutide, and mazdutide, which show promising weight-reducing effects [114]. The dual GLP-1/glucagon receptor agonist cotadutide and the triple GLP-1/GIP/glucagon receptor agonist efpeglenatide are being studied for the treatment of MAFLD [101]. The appetite-suppressing effect of pancreatic peptide YY (peptide tyrosine tyrosine) is well known, and initial clinical studies are underway with long-acting YY peptide analogs as monotherapy and in combination with GLP-1 receptor agonists for obesity treatment [115].

The amylin analog pramlintide, used in the United States as a prandial adjunct to insulin therapy, delays gastric emptying, improves postprandial blood sugar levels, and reduces weight [116]. A long-acting amylin analog similar to pramlintide, cagrilintide, administered parenterally once weekly at the highest dose of 4.5 mg, induced a significant 10.8% weight loss [117] (Figure 6). According to results from a recently published 32-week phase 2 trial, combined therapy with weekly parenteral administration of 2.4 mg cagrilintide and 2.4 mg semaglutide (CagriSema) reduced body weight by an average of 15.6% and HbA1c by an average of 2.2% in patients with T2DM compared to baseline [118] (Figure 7). Phase 3 trials with CagriSema combination therapy, part of the REDEFINE program, are evaluating its efficacy in weight loss and cardiovascular safety in patients with T2DM and overweight/obese, non-diabetic individuals [119]. The completion of the trials is expected in 2026 and 2027. Clinical trials have recently been initiated for obesity treatment with other long-acting amylin analogs and a dual amylin/calcitonin receptor agonist [120,121].

Further investigations are underway regarding protein-based approaches relying on other hormonal effects for effective weight loss; however, it currently appears that the potential clinical benefits of these are moderate. The appetite-suppressing effect of leptin analogs rapidly diminishes with the development of leptin resistance; thus, the use of metreleptin is limited to cases of congenital or acquired leptin deficiency [122]. Direct activation of the melanocortin-4 receptor in the hypothalamus with melanocortin-4 receptor agonist setmelanotide resulted in significant weight loss in small clinical studies conducted in severely obese individuals with pro-opiomelanocortin deficiency or other rare congenital conditions affecting the melanocortin-4 signaling pathway [123]. While inhibition of the appetite-stimulating (orexigenic) hormone ghrelin may seem an attractive therapeutic target, studies targeting ghrelin receptor inhibition have not reported sufficient efficacy to encourage further clinical development [124].

## 7. Limitations and Future Perspectives of Peptide-Based Therapy

Peptide-based therapies are emerging as promising treatments for diabetes and obesity due to their ability to effectively regulate glucose metabolism and body weight. However, the development and clinical application of these therapies will continue to face numerous challenges that must be addressed to optimize their efficacy and safety [125,126]. The process from peptide design and testing to the creation of a fully characterized therapeutic product is complex and fraught with challenges. Stability and degradation issues may arise during the development of peptide-based therapies, as peptides often have poor stability and are prone to rapid degradation, limiting their therapeutic potential. Immunological issues must also be considered; novel peptide structures can trigger immune responses, leading to potential immunological complications. Optimizing administration methods and ensuring targeted delivery to specific tissues remain significant obstacles. There is a risk of inducing severe hypoglycemia due to excessive lowering of blood sugar levels. Targeting multiple receptors with hybrid and chimeric peptides holds significant potential, but further research is needed to confirm their efficacy and safety. Targeting central and peripheral peptidergic systems, such as the central nervous system and the gastrointestinal tract, offers opportunities but also presents several challenges for clinical application and efficacy. Future research would focus on overcoming these barriers to fully realize the therapeutic benefits of peptides in the treatment of these conditions [125,126].

## 8. Conclusions

Obesity represents a serious public health issue, characterized as a chronic inflammatory condition closely associated with insulin resistance, T2DM, and an increased risk of cardiovascular diseases. Pharmacological treatment for obesity has seen dynamic development in recent years, with ongoing research focusing on the long-term clinical efficacy and safety of the employed formulations. These studies will further elucidate their role in the management of obesity and its related complications in the coming years.

Long-acting GLP-1 receptor agonists are widely applicable in the treatment of T2DM, known not only for their favorable effects on glycemic control and weight but also for significantly reducing the risk of cardiovascular diseases associated with atherosclerosis. GLP-1 receptor agonists have proven effective in the treatment of obesity and MAFLD alongside T2DM. Moreover, an increasing number of new and innovative formulations based on the mechanism of action of incretin hormones are becoming available for everyday clinical practice, including oral GLP-1 receptor agonists, the dual GLP-1/GIP receptor agonist tirzepatide, and other dual and triple GLP-1/GIP/glucagon receptor agonists, which may demonstrate further significant therapeutic potential. The dual GLP-1/GIP receptor agonist tirzepatide has already received approval in the United States and Europe, with the introduction of other dual receptor agonists expected soon. These novel agents may open further possibilities for future diabetes therapies in a personalized manner. For dual and triple GLP-1/GIP/glucagon receptor agonist formulations, as with efficacy, it is also warranted to assess safety through cardiovascular outcome trials.

## Figures and Tables

**Figure 1 biomedicines-12-01320-f001:**
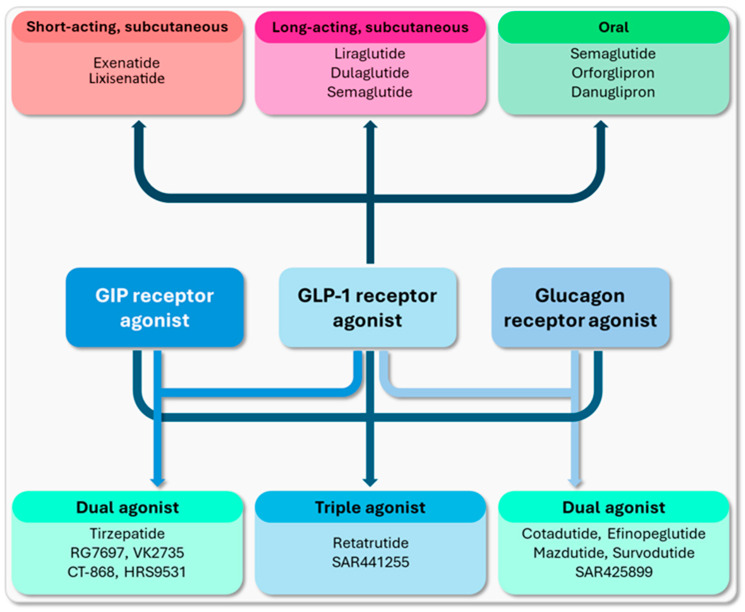
The incretin-based agents that are currently used in everyday medical practice or that are currently under clinical development [9].

**Figure 2 biomedicines-12-01320-f002:**
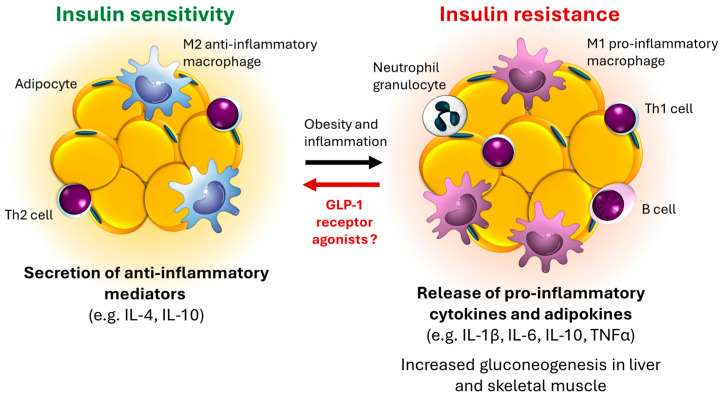
Inflammatory mediators in adipose tissue, contributing to the development of chronic inflammatory conditions [11]. GLP-1: glucagon-like peptide-1; TNF-α: tumor necrosis factor-alpha; Th1cell: T-helper 1; Th2 cell: T-helper 2 cell; IL-1β: interleukin-1β; IL-4: interleukin-4; IL-6: interleukin-6; IL-10: interleukin-10.

**Figure 3 biomedicines-12-01320-f003:**
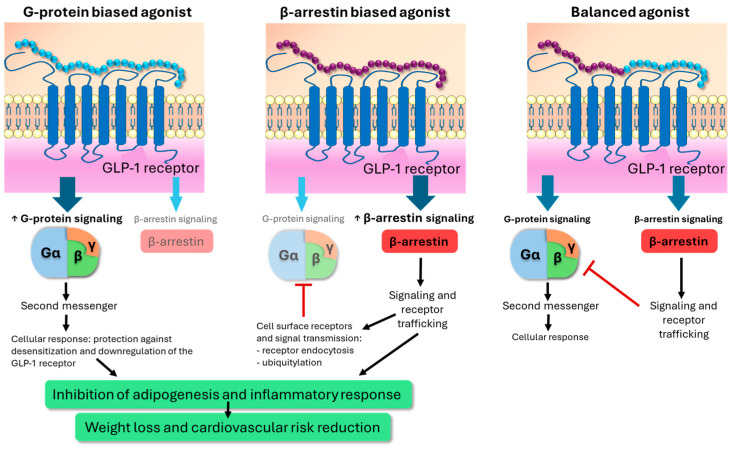
Activation of intracellular pathways by G-protein-coupled receptors and ligand binding, including G proteins, beta-arrestins, and other non-G-protein-regulated signaling pathways [20]. GLP-1: glucagon-like peptide-1.

**Figure 4 biomedicines-12-01320-f004:**
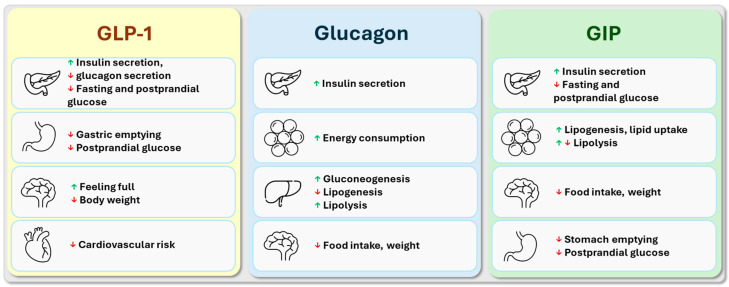
The main effects of glucagon-like peptide-1 (GLP-1), glucose-dependent insulinotropic polypeptide (GIP), and glucagon [9].

**Figure 5 biomedicines-12-01320-f005:**
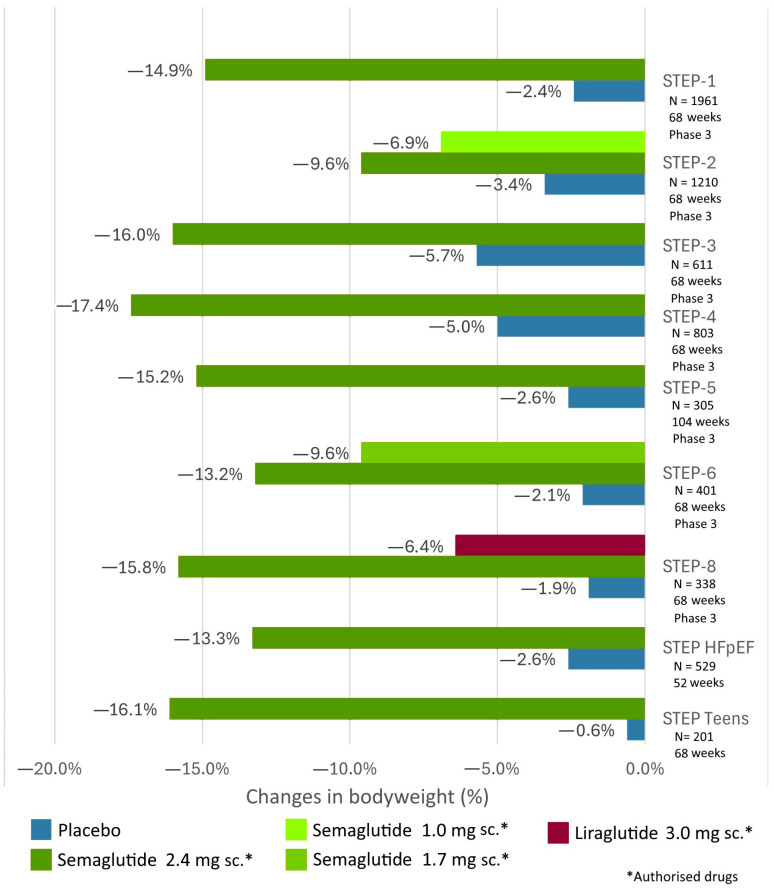
Clinical trials of subcutaneous semaglutide in overweight and obese individuals to evaluate weight loss efficacy (STEP 1–6, STEP-8, STEP Teens, STEP HFpEF).

**Figure 6 biomedicines-12-01320-f006:**
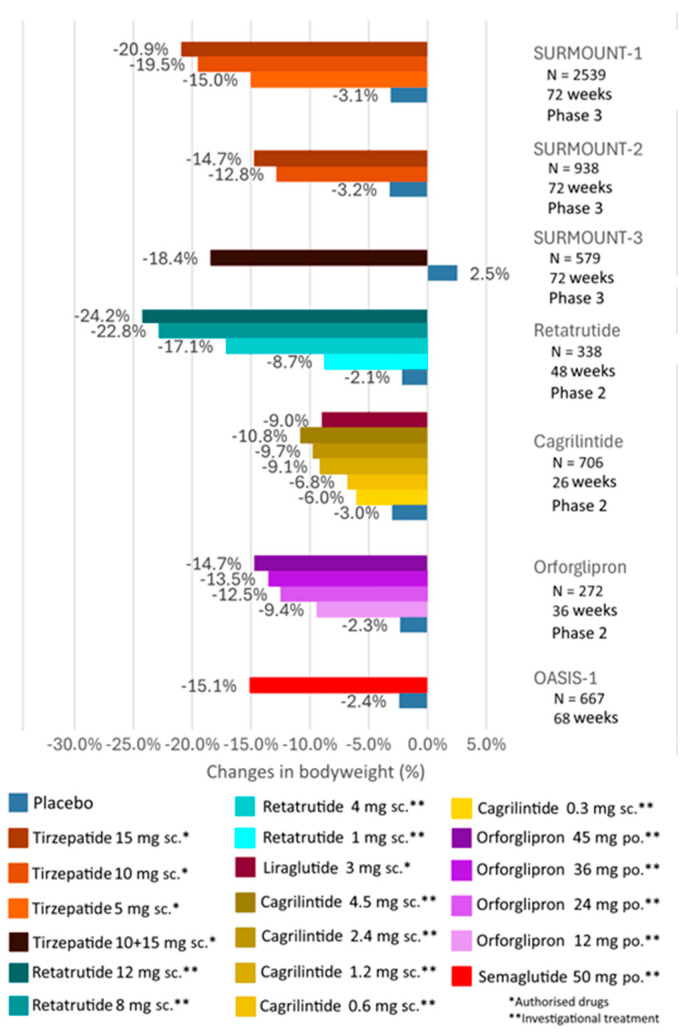
Clinical trials of subcutaneous tirzepatide (SURMOUNT-1, -2 and -3 trials), retatrutide and cagrilintide, and oral semaglutide and orforglipron in overweight and obese subjects to evaluate weight reduction efficacy.

**Figure 7 biomedicines-12-01320-f007:**
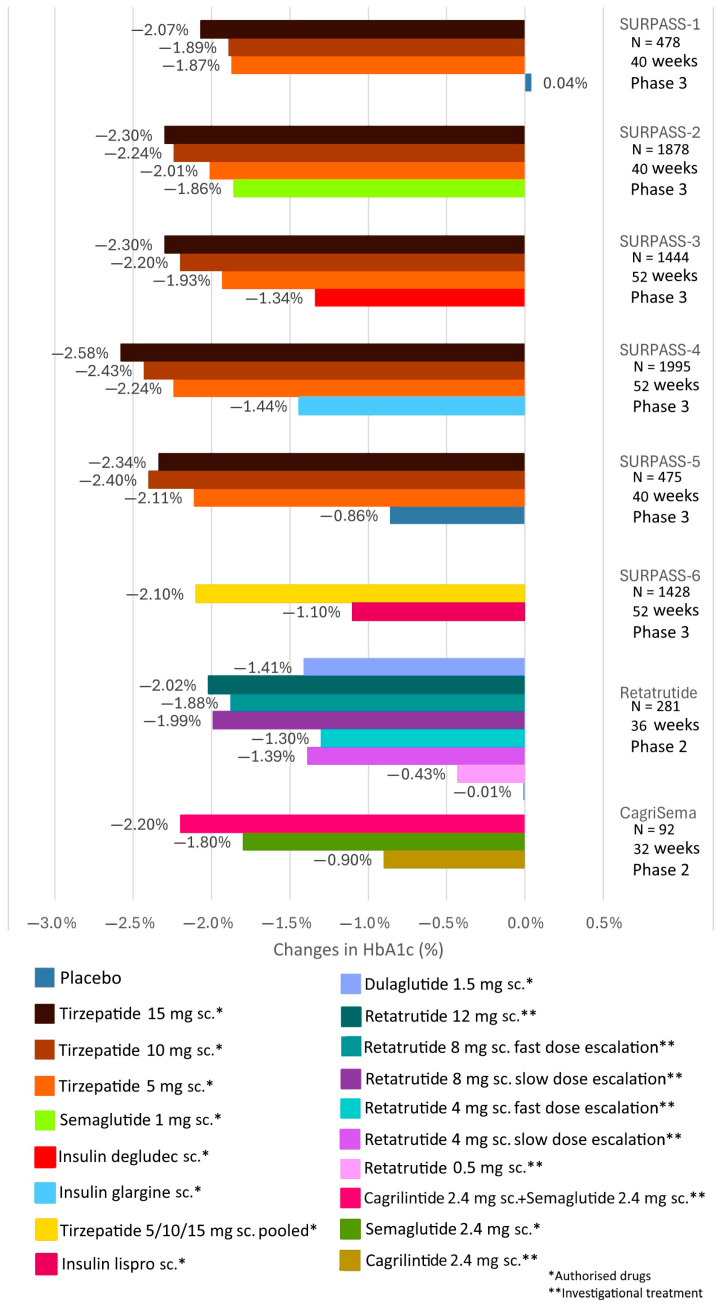
Clinical trials of subcutaneous tirzepatide, retatrutide, and cagrilintide/semaglutide in type 2 diabetic subjects to evaluate HbA1c reduction efficacy.

## Data Availability

All data generated or analyzed during this study are included in this published article.

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
