# Peer review of "New Developments in Pharmacological Treatment of Obesity and Type 2 Diabetes—Beyond and within GLP-1 Receptor Agonists"

_biomedicines, 2024, doi:10.3390/biomedicines12061320_

Round 1
Reviewer 1 Report
Comments and Suggestions for Authors
I read with great interest the paper “New developments in pharmacological treatment of obesity and type 2 diabetes – Beyond and within GLP-1 receptor agonists”.
English is generally sound, only minor revision.
1. The introduction is well-structured, with clear subsections addressing the significance of weight loss in T2DM development, the importance of lifestyle changes, the emergence of incretin-based therapies, and the role of inflammatory processes in adipose tissue. However, the transitions between these subsections could be smoother to improve the overall flow of the text.
2. While the text discusses the effects of GIP on lipid metabolism and atherosclerosis, it acknowledges that the precise effects are unclear and may vary based on different studies. Providing a more nuanced discussion of conflicting findings and areas needing further research could enhance the section on GIP.
3. The text briefly mentions the side effects associated with peptide-based therapies, such as nausea and vomiting with high-dose semaglutide. Expanding on the safety profile, including other potential adverse effects and long-term considerations, could provide a more comprehensive understanding of these treatments.
4. While the text provides a detailed summary of the development and efficacy of peptide-based therapies, it lacks critical analysis or discussion of potential limitations, challenges, or areas for future research. Including such analysis would provide a more balanced perspective on the topic.
5. As inflammation, oxidative stress and type 2 diabetes complications are tightly associated, you can provide a text explaining this association (doi: 10.3390/cimb45080420), and enrich it with data of current studies if available.
Comments on the Quality of English LanguageMinor, some sentences should be improved for readability
Author Response
Thank you for the thoughtful review and the positive comments on our manuscript.
- The introduction is well-structured, with clear subsections addressing the significance of weight loss in T2DM development, the importance of lifestyle changes, the emergence of incretin-based therapies, and the role of inflammatory processes in adipose tissue. However, the transitions between these subsections could be smoother to improve the overall flow of the text.
Response: The Reviewer is absolutely correct, we modified the "Introduction" section as follows: “A therapeutic approach based on innovation and the discovery of mechanisms of action offers promising opportunities for a deeper understanding of the connections between obesity and T2DM, which may lead to the development of new therapeutic options in addition to lifestyle changes” (chapter 1, page 2) and “Clinical trials concluded in the past decades have confirmed that GLP-1 receptor agonists have proven effective in the management of T2DM, and some formulations are now also used in higher doses for the treatment of obesity” (chapter 1, page 2)
- While the text discusses the effects of GIP on lipid metabolism and atherosclerosis, it acknowledges that the precise effects are unclear and may vary based on different studies. Providing a more nuanced discussion of conflicting findings and areas needing further research could enhance the section on GIP.
Response: We agree with the Reviewer, we modified the following sentence to “The Main Effects of Glucose-Dependent Insulinotropic Polypeptide” section: “The anti-inflammatory effect of GIP on atherosclerosis is not clear from preclinical studies, therefore the role of GIP in the atherosclerosis process seems to be controversial at the moment.” (chapter 3.2, page 8)
- The text briefly mentions the side effects associated with peptide-based therapies, such as nausea and vomiting with high-dose semaglutide. Expanding on the safety profile, including other potential adverse effects and long-term considerations, could provide a more comprehensive understanding of these treatments.
Response: We completely agree with the Reviewer, we added the following sentences to “Peptide-based therapies in the treatment of obesity” section: “High-dose semaglutide and tirzepatide have been extensively studied for their efficacy in weight management and glycemic control. However, their use can be associated with several side effects, which are important to consider during clinical application. Gastrointestinal symptoms such as nausea, vomiting, diarrhea, or constipation were the most common side effects, observed at a significantly higher incidence in patients receiving high-dose semaglutide or tirzepatide compared to comparators. These side effects are usually mild to moderate in severity and tend to be temporary. The use of semaglutide and tirzepatide may increase the risk of biliary disorders such as cholelithiasis, a risk similar to that of other GLP-1 receptor agonists, and therefore patients should be monitored during clinical use (Bradley 2022, Mishra 2023). There is concern about the potential risk of pancreatitis and pancreatic and thyroid cancers with the use of GLP-1 receptor agonists. However, the incidence of these conditions is low, and definitive conclusions cannot be drawn at this time. In clinical trials, severe hypoglycemia, acute pancreatitis, gallstones, and cholecystitis occurred rarely and were observed only in a few cases during the use of high-dose semaglutide and tirzepatide (Bradley 2022, Mishra 2023).” (chapter 4, page 12)
Ref: Bradley CL, McMillin SM, Hwang AY, Sherrill CH. High-Dose Once-Weekly Semaglutide: A New Option for Obesity Management. Ann Pharmacother. 2022 Aug;56(8):941-950. doi: 10.1177/10600280211053867.
Mishra R, Raj R, Elshimy G, Zapata I, Kannan L, Majety P, Edem D, Correa R. Adverse Events Related to Tirzepatide. J Endocr Soc. 2023 Jan 26;7(4):bvad016. doi: 10.1210/jendso/bvad016.
- While the text provides a detailed summary of the development and efficacy of peptide-based therapies, it lacks critical analysis or discussion of potential limitations, challenges, or areas for future research. Including such analysis would provide a more balanced perspective on the topic.
Response: In accordance with the reviewer's request, we supplemented the manuscript and added a new chapter: “Chapter 7: Limitations and future perspectives of peptide-based therapy”, which contains the following: “Peptide-based therapies are emerging as promising treatments for diabetes and obesity due to their ability to effectively regulate glucose metabolism and body weight. However, the development and clinical application of these therapies will continue to face numerous challenges that must be addressed to optimize their efficacy and safety (Bailey 2018, Casey 2021). The process from peptide design and testing to the creation of a fully characterized therapeutic product is complex and fraught with challenges. Stability and degradation issues may arise during the development of peptide-based therapies, as peptides often have poor stability and are prone to rapid degradation, limiting their therapeutic potential. Immunological issues must also be considered; novel peptide structures can trigger immune responses, leading to potential immunological complications. Optimizing administration methods and ensuring targeted delivery to specific tissues remain significant obstacles. There is a risk of inducing severe hypoglycemia due to excessive lowering of blood sugar levels. Targeting multiple receptors with hybrid and chimeric peptides holds significant potential, but further research is needed to confirm their efficacy and safety. Targeting central and peripheral peptidergic systems, such as the central nervous system and the gastrointestinal tract, offers opportunities but also presents several challenges for clinical application and efficacy. Future research would focus on overcoming these barriers to fully realize the therapeutic benefits of peptides in the treatment of these conditions (Bailey 2018, Casey 2021).” (chapter 7, page 17)
Ref: Bailey CJ. Glucose-lowering therapies in type 2 diabetes: Opportunities and challenges for peptides. Peptides. 2018 Feb;100:9-17. doi: 10.1016/j.peptides.2017.11.012.
Casey R, Adelfio A, Connolly M, Wall A, Holyer I, Khaldi N. Discovery through Machine Learning and Preclinical Validation of Novel Anti-Diabetic Peptides. Biomedicines. 2021 Mar 9;9(3):276. doi: 10.3390/biomedicines9030276.
- 5.As inflammation, oxidative stress and type 2 diabetes complications are tightly associated, you can provide a text explaining this association (doi: 10.3390/cimb45080420), and enrich it with data of current studies if available.
Response: We agree with the Reviewer, we added the following sentence to the manuscript: “Obesity and associated insulin resistance are accompanied by increased oxidative stress in endothelial cells, during which the production of reactive oxygen species (ROS) surpasses the capacity of antioxidant factors to balance them. As a result, signaling pathways are modified, and the lipid, protein, and DNA content of cells is damaged. The superoxide anion, with high reactivity, binds and inactivates the physiological function of endothelial nitric oxide synthase (eNOS), leading to the production of peroxynitrite instead of NO. Peroxynitrite is a potent oxidizing agent that can cause lipid peroxidation, protein tyrosine nitration, DNA damage, and cell death. Additionally, peroxynitrite oxidizes tetrahydrobiopterin, an essential cofactor for eNOS, rendering it biologically inactive and unable to catalyze endothelial NO synthesis (Caturano 2023). Consequently, eNOS catalyzes the production of further superoxide radicals instead of NO. Beyond other intracellular signaling and metabolic changes, the inactivation of eNOS is a key factor in the relationship between oxidative stress and pathological functional changes in the endothelium. In diabetes, chronic hyperglycemia, acute fluctuations in blood glucose levels, and insulin resistance all contribute to increased oxidative stress and, thereby, to the development of endothelial dysfunction (Meza 2019).” (chapter 2, page 4) We have added the recommended literature reference to the "References" section (Ref 15)
Ref: Caturano A, D’Angelo M, Mormone A, Russo V, Mollica MP, Salvatore T, Galiero R, Rinaldi L, Vetrano E, Marfella R, et al. Oxidative Stress in Type 2 Diabetes: Impacts from Pathogenesis to Lifestyle Modifications. Current Issues in Molecular Biology. 2023; 45(8):6651-6666. https://doi.org/10.3390/cimb45080420
Meza CA, La Favor JD, Kim DH, et al.: Endothelial Dysfunction: Is There a Hyperglycemia-Induced Imbalance of NOX and NOS? Int J Mol Sci 2019; 20(15): 3775. doi: 10.3390/ijms20153775.
Thank you again for your thorough review of our manuscript and your valuable feedback!
Reviewer 2 Report
Comments and Suggestions for Authors
The manuscript is interesting and quite well written. It is well structured and updated. The figures are clear and functional for understanding the text.
This reviewer raises some comments that could enrich the manuscript.
The weight loss effect in obese and overweight subjects brought about by GLP-1-RA and other drugs discussed in the manuscript can achieve an important reduction in cardiac insulin resistance which is related to a reduced ejection fraction, as it has been demonstrated both in diabetics (J Am Coll Cardiol. 2000 Jul;36(1):219-26 doi: 10.1016/s0735-1097(00)00717-8) and in simply obese non-diabetic patients (Eur Heart J. 2005 Jun; 26(12): 1205-12doi: 10.1093/eurheartj/ehi271.). These effects could partly justify the important cardioprotective action of these drugs against weight loss, in particular against heart failure. This important issue should be addressed in the text.
GLP-1 RAs may play other important, less well-known health roles with respect to cardio-nephroprotective and anti-obesity effects, not only in the diabetic population. In particular, GLP1-RAs have protective effects on the microvascular compartment of the myocardium (Biomedicines. 2022 Sep 14;10(9):2274. doi: 10.3390/biomedicines10092274.). Furthermore, numerous evidence is currently available on the role of GLP-1 RAs in the treatment of epicardial adipose tissue (Biomolecules. January 21, 2022;12(2):176. doi: 10.3390/biom12020176.) as well as the NAFLD (Int J Mol Sci. 2023 Jan 15;24(2):1703. doi: 10.3390/ijms24021703.) and to hypothesize potential future scenarios. These actions of GLP-1 should be discussed in the text.
Comments on the Quality of English LanguageMinor editing of English language is required.
Author Response
Thank you for the thoughtful review and the positive comments on our manuscript.
- The weight loss effect in obese and overweight subjects brought about by GLP-1-RA and other drugs discussed in the manuscript can achieve an important reduction in cardiac insulin resistance which is related to a reduced ejection fraction, as it has been demonstrated both in diabetics (J Am Coll Cardiol. 2000 Jul;36(1):219-26 doi: 10.1016/s0735-1097(00)00717-8) and in simply obese non-diabetic patients (Eur Heart J. 2005 Jun; 26(12): 1205-12doi: 10.1093/eurheartj/ehi271.). These effects could partly justify the important cardioprotective action of these drugs against weight loss, in particular against heart failure. This important issue should be addressed in the text.
Response: At the request of the Reviewer, the following sentences were added to the section "The Main Effects of Glucagon-Like Peptide-1": „The weight-reducing effect achieved with GLP-1 receptor agonists may alleviate cardiac insulin resistance in obese and overweight individuals. Reduced insulin resistance may be linked to decreased ejection fraction in individuals with T2DM (Sasso 2000) and in non-diabetic obese patients (Sasso 2005). These effects may partially contribute to the cardioprotective effects of agents targeting the incretin axis, particularly in the context of heart failure.” (chapter 3.1, page 7) We have added the recommended literature reference to the "References" section (Ref 44 and 45)
Ref: Sasso FC, Carbonara O, Cozzolino D, Rambaldi P, Mansi L, Torella D, Gentile S, Turco S, Torella R, Salvatore T. Effects of insulin-glucose infusion on left ventricular function at rest and during dynamic exercise in healthy subjects and noninsulin dependent diabetic patients: a radionuclide ventriculographic study. J Am Coll Cardiol. 2000 Jul;36(1):219-26. doi: 10.1016/s0735-1097(00)00717-8.
Sasso FC, Carbonara O, Nasti R, Marfella R, Esposito K, Rambaldi P, Mansi L, Salvatore T, Torella R, Cozzolino D. Effects of insulin on left ventricular function during dynamic exercise in overweight and obese subjects. Eur Heart J. 2005 Jun;26(12):1205-12. doi: 10.1093/eurheartj/ehi271.
- GLP-1 RAs may play other important, less well-known health roles with respect to cardio-nephroprotective and anti-obesity effects, not only in the diabetic population. In particular, GLP1-RAs have protective effects on the microvascular compartment of the myocardium (Biomedicines. 2022 Sep 14;10(9):2274. doi: 10.3390/biomedicines10092274.). Furthermore, numerous evidence is currently available on the role of GLP-1 RAs in the treatment of epicardial adipose tissue (Biomolecules. January 21, 2022;12(2):176. doi: 10.3390/biom12020176.) as well as the NAFLD (Int J Mol Sci. 2023 Jan 15;24(2):1703. doi: 10.3390/ijms24021703.) and to hypothesize potential future scenarios. These actions of GLP-1 should be discussed in the text.
Response: In agreement with the reviewer, we inserted the following sentences in the section "Chapter 3.1.: The Main Effects of Glucagon-Like Peptide-1": „Furthermore, GLP-1 receptor agonists have been demonstrated to exert a protective effect on the microvascular compartment of the myocardium (Salvatore 2022). Substantial evi-dence also supports the beneficial effects of the GLP-1 hormone on epicardial adipose tis-sue (Salvatore 2022) and in the treatment of metabolic associated fatty liver disease (MAFLD) (Nevola 2023).” (chapter 3, page 7-8) We have added the recommended literature reference to the "References" section (Ref 47 and 48 and 49)
Ref: Salvatore T, Galiero R, Caturano A, Vetrano E, Loffredo G, Rinaldi L, Catalini C, Gjeloshi K, Albanese G, Di Martino A, Docimo G, Sardu C, Marfella R, Sasso FC. Coronary Microvascular Dysfunction in Diabetes Mellitus: Pathogenetic Mechanisms and Potential Therapeutic Options. Biomedicines. 2022 Sep 14;10(9):2274. doi: 10.3390/biomedicines10092274.
Salvatore T, Galiero R, Caturano A, Vetrano E, Rinaldi L, Coviello F, Di Martino A, Albanese G, Colantuoni S, Medicamento G, Marfella R, Sardu C, Sasso FC. Dysregulated Epicardial Adipose Tissue as a Risk Factor and Potential Therapeutic Target of Heart Failure with Preserved Ejection Fraction in Diabetes. Biomolecules. 2022 Jan 21;12(2):176. doi: 10.3390/biom12020176.
Nevola R, Epifani R, Imbriani S, Tortorella G, Aprea C, Galiero R, Rinaldi L, Marfella R, Sasso FC. GLP-1 Receptor Agonists in Non-Alcoholic Fatty Liver Disease: Current Evidence and Future Perspectives. Int J Mol Sci. 2023 Jan 15;24(2):1703. doi: 10.3390/ijms24021703.
Thank you again for your thorough review of our manuscript and your valuable feedback!
Reviewer 3 Report
Comments and Suggestions for Authors
This is a correct review but bibliography could be improved
Please add
Cells
. 2023 Dec 28;13(1):65. doi: 10.3390/cells13010065.
GLP1 Receptor Agonists-Effects beyond Obesity and Diabetes
GLP-1 physiology informs the pharmacotherapy of obesity.
Drucker DJ.
Mol Metab. 2022 Mar;57:101351. doi: 10.1016/j.molmet.2021.101351. Epub 2021 Oct 6.
PMID: 34626851 Free PMC article. Review.
Differential cardiovascular and renal benefits of SGLT2 inhibitors and GLP1 receptor agonists in patients with type 2 diabetes mellitus.
Kim CH, Hwang IC, Choi HM, Ahn CH, Yoon YE, Cho GY.
Int J Cardiol. 2022 Oct 1;364:104-111. doi: 10.1016/j.ijcard.2022.06.027. Epub 2022 Jun 15.
PMID: 35716949
Comparing benefits from sodium-glucose cotransporter-2 inhibitors and glucagon-like peptide-1 receptor agonists in randomized clinical trials: a network meta-analysis.
Sabouret P, Bocchino PP, Angelini F, D'Ascenzo F, Galati G, Fysekidis M, DE Ferrari GM, Fischman DL, Bhatt DL, Biondi-Zoccai G.
Minerva Cardiol Angiol. 2023 Apr;71(2):199-207. doi: 10.23736/S2724-5683.22.05900-2. Epub 2022 Feb 23.
PMID: 35195376
Expected Health Benefits of SGLT-2 Inhibitors and GLP-1 Receptor Agonists in Older Adults.
Dadwani RS, Wan W, Skandari MR, Huang ES.
MDM Policy Pract. 2023 Jul 20;8(2):23814683231187566. doi: 10.1177/23814683231187566. eCollection 2023 Jul-Dec.
PMID: 37492502 Free PMC article.
Beyond glucose lowering: glucagon-like peptide-1 receptor agonists, body weight and the cardiovascular system.
Vergès B, Bonnard C, Renard E.
Diabetes Metab. 2011 Dec;37(6):477-88. doi: 10.1016/j.diabet.2011.07.001. Epub 2011 Aug 25.
PMID: 21871831 Review.
Author Response
Thank you for the review and the positive comments on our manuscript.
Please add:
- Wilbon SS, Kolonin MG. GLP1 Receptor Agonists-Effects beyond Obesity and Diabetes. Cells. 2023 Dec 28;13(1):65. doi: 10.3390/cells13010065. PMID: 38201269; PMCID: PMC10778154.
- Drucker DJ. GLP-1 physiology informs the pharmacotherapy of obesity. Mol Metab. 2022 Mar;57:101351. doi: 10.1016/j.molmet.2021.101351.
- Kim CH, Hwang IC, Choi HM, Ahn CH, Yoon YE, Cho GY. Differential cardiovascular and renal benefits of SGLT2 inhibitors and GLP1 receptor agonists in patients with type 2 diabetes mellitus. Int J Cardiol. 2022 Oct 1;364:104-111. doi: 10.1016/j.ijcard.2022.06.027.
- Sabouret P, Bocchino PP, Angelini F, D'Ascenzo F, Galati G, Fysekidis M, DE Ferrari GM, Fischman DL, Bhatt DL, Biondi-Zoccai G. Comparing benefits from sodium-glucose cotransporter-2 inhibitors and glucagon-like peptide-1 receptor agonists in randomized clinical trials: a network meta-analysis. Minerva Cardiol Angiol. 2023 Apr;71(2):199-207. doi: 10.23736/S2724-5683.22.05900-2.
- Dadwani RS, Wan W, Skandari MR, Huang ES. Expected Health Benefits of SGLT-2 Inhibitors and GLP-1 Receptor Agonists in Older Adults. MDM Policy Pract. 2023 Jul 20;8(2):23814683231187566. doi: 10.1177/23814683231187566.
- Vergès B, Bonnard C, Renard E. Beyond glucose lowering: glucagon-like peptide-1 receptor agonists, body weight and the cardiovascular system. Diabetes Metab. 2011 Dec;37(6):477-88. doi: 10.1016/j.diabet.2011.07.001.
Response: In accordance with the request of the reviewer, we cited the requested literature in the manuscript (Ref 32, 36, 39, 41, 43, 46)
Thank you again for your review of our manuscript!
Reviewer 4 Report
Comments and Suggestions for Authors
The topic of this manuscript is interesting, aiming to provide an update on the main data present in the scientific stream regarding the use of newly developed molecules in the treatment of obesity and type 2 diabetes. The manuscript is well written, but some aspects could be improved:
· Please mention the type of review in the Introduction section.
· It would be better to present the prevalence of diabetes and obesity and the negative impact of these pathologies on health status.
· Please present the search strategy (criteria) of information or references.
· Authors should present the difference between STEPs trials. Also, for SURMOUNT-1, -2 and -3 trials
· Figure 1: please use caps look for “lixisenatide”.
· Figure 6 and 7. Please use different colours; the current ones may create confusion. You could split the figures into two: (a) authorised drugs; (b) investigational treatment.
· The authors should also refer to the adverse reactions of these drugs, especially those that can limit their use (pancreatitis, kidney injury, etc.). Also, the results of some pharmacovigilance studies from the most important databases (EudraVigilance, VigiBase, FAERS, etc.) should be present.
Author Response
Thank you for the thoughtful review and the positive comments on our manuscript.
- Please mention the type of review in the Introduction section.
Response: We have modified the abstract and the introduction as follows: „This narrative review summarizes the therapeutic effects of different incretin hormones and to present future prospects in the treatment of T2DM and obesity.” (Abstract, page 1) and „The purpose of this narrative review is to summarize the therapeutic effects of various in-cretin hormones and present future prospects in the treatment of T2DM and obesity.” (Introduction, page 3, paragraph 1)
- It would be better to present the prevalence of diabetes and obesity and the negative impact of these pathologies on health status.
Response: We completely agree with the Reviewer, we added the following sentences to “Introduction” section: „The prevalence of diabetes and obesity has been rising globally, posing significant public health challenges. According to an analysis of global data, adult obesity rates have more than doubled in women and nearly tripled in men. In 2022, a total of 879 million adults worldwide lived with obesity. Obesity is responsible for approximately 43% of all cases of type 2 diabetes, with rates as high as 80-85% in some populations (NCD Risk Factor Collaboration 2024). Obesity not only increases the risk of developing type 2 diabe-tes but also exacerbates complications and related comorbidities such as cardiovascular disease, cancer, and osteoarthritis, while decreasing life expectancy and raising health care costs.” (Introduction, page 1, paragraph 1)
Ref: NCD Risk Factor Collaboration (NCD-RisC). Worldwide trends in underweight and obesity from 1990 to 2022: a pooled analysis of 3663 population-representative studies with 222 million children, adolescents, and adults. Lancet. 2024 Mar 16;403(10431):1027-1050. doi: 10.1016/S0140-6736(23)02750-2.
- Please present the search strategy (criteria) of information or references.
Response: We added the following sentences to “Introduction” section: “We obtained the data for this review from PubMed, Scopus, Web of Science, and Clinical-Trials.gov, using the following keywords: "obesity", "type 2 diabetes", "new drug treat-ment", "recent developments", "clinical studies", "G protein-coupled receptor", "gluca-gon-like peptide-1 receptor agonists", "glucose-dependent insulinotropic polypeptide re-ceptor agonist", "glucagon receptor agonist", "dual agonist", and "triple agonist".” (Introduction, page 3, paragraph 1)
- Authors should present the difference between STEPs trials. Also, for SURMOUNT-1, -2 and -3 trials
Response: We added the following sentences to “chapter 4. Peptide-based therapies in the treatment of obesity” section: „The STEP clinical trials collectively provide valuable information about semaglutide’s safety, efficacy, and impact on weight loss and cardiovascular outcomes in various patient populations.” (page 10) and „The SURMOUNT trials have demonstrated tirzepatide glucose-lowering efficacy, weight loss efficacy, and cardiovascular safety in various patient populations.” (page 12)
- Figure 1: please use caps look for “lixisenatide”.
Response: We have made the requested modification (Fig 1).
- Figure 6 and 7. Please use different colours; the current ones may create confusion. You could split the figures into two: (a) authorised drugs; (b) investigational treatment.
Response: We have made the requested modification (Fig 6 and 7).
- The authors should also refer to the adverse reactions of these drugs, especially those that can limit their use (pancreatitis, kidney injury, etc.). Also, the results of some pharmacovigilance studies from the most important databases (EudraVigilance, VigiBase, FAERS, etc.) should be present.
Response: We completely agree with the Reviewer, we added the following sentences to “Peptide-based therapies in the treatment of obesity” section: “High-dose semaglutide and tirzepatide have been extensively studied for their efficacy in weight management and glycemic control. However, their use can be associated with several side effects, which are important to consider during clinical application. Gastrointestinal symptoms such as nausea, vomiting, diarrhea, or constipation were the most common side effects, observed at a significantly higher incidence in patients receiving high-dose semaglutide or tirzepatide compared to comparators. These side effects are usually mild to moderate in severity and tend to be temporary. The use of semaglutide and tirzepatide may increase the risk of biliary disorders such as cholelithiasis, a risk similar to that of other GLP-1 receptor agonists, and therefore patients should be monitored during clinical use (Bradley 2022, Mishra 2023). There is concern about the potential risk of pancreatitis and pancreatic and thyroid cancers with the use of GLP-1 receptor agonists. However, the incidence of these conditions is low, and definitive conclusions cannot be drawn at this time. In clinical trials, severe hypoglycemia, acute pancreatitis, gallstones, and cholecystitis occurred rarely and were observed only in a few cases during the use of high-dose semaglutide and tirzepatide (Bradley 2022, Mishra 2023).” (chapter 4, page 12)
Ref: Bradley CL, McMillin SM, Hwang AY, Sherrill CH. High-Dose Once-Weekly Semaglutide: A New Option for Obesity Management. Ann Pharmacother. 2022 Aug;56(8):941-950. doi: 10.1177/10600280211053867.
Mishra R, Raj R, Elshimy G, Zapata I, Kannan L, Majety P, Edem D, Correa R. Adverse Events Related to Tirzepatide. J Endocr Soc. 2023 Jan 26;7(4):bvad016. doi: 10.1210/jendso/bvad016.
Thank you again for your thorough review of our manuscript and your valuable feedback!
Round 2
Reviewer 1 Report
Comments and Suggestions for Authors
The authors addressed all issues raised by this reviewer. No further comments.
Reviewer 3 Report
Comments and Suggestions for Authors
The new review ajusts all changes beg about
Congratulations!